# Comparison of the Visibility of Canine Menisci before and after Tibial Plateau Leveling Osteotomy: 3D-Printed Model Study

**DOI:** 10.3390/ani14010065

**Published:** 2023-12-23

**Authors:** Piotr Trębacz, Jan Frymus, Mateusz Pawlik, Anna Barteczko, Aleksandra Kurkowska, Michał Czopowicz, Magdalena Antonowicz, Wojciech Kajzer

**Affiliations:** 1Department of Surgery and Anesthesiology of Small Animals, Institute of Veterinary Medicine, Warsaw University of Life Sciences-SGGW, Nowoursynowska 159c, 02-776 Warsaw, Poland; 2CABIOMEDE Ltd., Karola Olszewskiego 21, 25-663 Kielce, Poland; mateusz.pawlik@cabiomede.com (M.P.); anna.barteczko@cabiomede.com (A.B.); aleksandra.kurkowska@cabiomede.com (A.K.); 3Division of Veterinary Epidemiology and Economics, Institute of Veterinary Medicine, Warsaw University of Life Sciences-SGGW, Nowoursynowska 159c, 02-776 Warsaw, Poland; michal_czopowicz@sggw.edu.pl; 4Department of Biomaterials and Medical Devices Engineering, Faculty of Biomedical Engineering, Silesian University of Technology, Roosevelta 40, 41-800 Zabrze, Poland; magdalena.antonowicz@polsl.pl (M.A.); wojciech.kajzer@polsl.pl (W.K.)

**Keywords:** cranial cruciate ligament disease, dog, laser scanning, reverse engineering, additive manufacturing, proximal tibial osteotomy

## Abstract

**Simple Summary:**

Cranial cruciate ligament (CCL) insufficiency is a common condition in dogs, commonly treated using tibial plateau leveling osteotomy (TPLO). CCL injury often results in secondary abnormalities of the menisci. Arthrotomy is still used by many veterinarians to detect meniscal damage, and also, before TPLO an assessment of the menisci is generally recommended. However, only a limited part of the menisci is visible during arthrotomy of the stifle. Our hypothesis was that the change in tibial plateau position created by TPLO allows a better visual access to the menisci. To verify suspicion, we compared the degree of visibility of the lateral and medial menisci before and after TPLO on 15 identical 3-dimensional (3D)-printed models of a normal canine tibia. On each model, the meniscal area was stained and photographed, and its part visible to the surgeon before and after TPLO was digitally measured and compared. Our modeling showed that before TPLO, without additional instrumentation, e.g., arthroscopy, only approximately 16% of the entire meniscus area was visible. We demonstrated that TPLO increased the visibility of the lateral meniscus to 38–56% (mean ± SD: 46.5 ± 5.4%) of its entire area and of the medial meniscus to 41–70% (mean ± SD: 52.8 ± 7.6%). This increase in the visibility was statistically significant (*p* < 0.001). We conclude that performing the examination of the menisci after TPLO allows for a review of about 50% of the entire meniscal area, in contrast to examination before TPLO, where only about 16% is visible. The visible area is slightly smaller in the lateral than in the medial meniscus; however, this difference is unlikely to be clinically relevant.

**Abstract:**

The aim of this study was to compare the degree of visibility of the lateral and medial menisci before and after tibial plateau leveling osteotomy (TPLO) on 3D-printed models created after laser scanning of the right tibia with menisci derived from a fresh cadaver of a 4-year-old adult male golden retriever. The models were produced of white polylactic acid, and the menisci were filled with light-curing red resin. The models showed a similar conformation as the natural specimen harvested from the cadaver, maintaining the same length and width, in addition to reproducing the anatomical structures. From the pre- and post-TPLO radiographs, it was possible to identify the anatomical structures corresponding to the tibial plateau. The preoperative tibial plateau angle was 26.2°, and the postoperative one ranged between 4.0° and 5.3° (4.6 ± 0.4°). In the bird’s-eye photo, the total number of red pixels in the lateral and the medial meniscus was 2,053,995 and 2,140,939, respectively. Before TPLO, only between 14% and 19% of the entire area of the menisci was visible, and the unhidden part of the entire area of the meniscus before TPLO did not differ significantly between the lateral (16.2 ± 1.6%) and the medial (16.4 ± 1.6%) meniscus (*p* = 0.351). The visible part of the entire meniscus area increased significantly after TPLO both in the lateral and medial menisci (*p* < 0.001)—mean difference ± SD of 30.3 ± 4.3% (CI 95%: 27.9%, 32.6%) and 36.4 ± 6.4% (CI 95%: 32.9%, 40.0%), respectively. In conclusion, the intraoperative examination and treatment of dog menisci are easier after TPLO.

## 1. Introduction

Meniscal injury is a common sequela of cranial cruciate ligament (CCL) insufficiency in dogs, with the reported prevalence of medial meniscal injury ranging from 10% to 80% [1]. The wide range of reported prevalence is likely due to the varying sensitivity of the different surgical approaches (i.e., arthrotomy vs. arthroscopy) and a surgeon’s experience.

Arthroscopy is recognized as the gold standard for the diagnosis of meniscal tears because it magnifies and illuminates the stifle joint, allowing a better assessment of intra-articular structures compared to arthrotomy. However, arthrotomy, e.g., mini-arthrotomy, is still used by many veterinarians due to arthroscopic equipment limitations, a steep learning curve of arthroscopy, and financial concerns.

The anatomy of the proximal tibia is complex. The articular surface is formed by the lateral and medial condyles. Structures associated with the proximal tibia include the patellar ligament, the lateral and medial menisci, the tibial portions of the cranial and caudal cruciate ligaments, the intermeniscal ligament, the lateral collateral ligament, the medial collateral ligament, and the popliteal tendon of origin. In addition, the femorotibial joint space is narrow in the dogs, making accurate visualization and treatment of the menisci challenging. Otherwise, there is a risk of iatrogenic cartilage damage when inserting instruments during surgery due to the limited space. Intra- and extra-articular stifle distraction has been advocated as a strategy to improve exposure and working space to avoid these injuries [1,2,3,4].

The majority of meniscal injuries involve the caudal pole of the medial meniscus, which is hardly visible during arthrotomy. This requires strategies to retrieve and resect the pathological meniscal tissue, which is often displaced to the most caudal aspect of the joint.

Tibial plateau leveling osteotomy (TPLO) remains one of the most commonly performed techniques to treat CCL insufficiency. Following a radial osteotomy of the proximal epiphysis, the plateau is leveled to an angle of approximately 5°. The change in tibial plateau position created by TPLO also changes the viewing angle of the tibial plateau, allowing a different perspective and different access to the menisci, particularly the caudal horn of the medial meniscus.

To the authors’ knowledge, a comparison of the degree of visibility of the menisci before and after semicircular osteotomy of the tibia has not been analyzed. The aim of this study was to compare the degree of visibility of the lateral and medial menisci before and after TPLO on a 3D-printed model of the dog tibia.

## 2. Materials and Methods

### 2.1. 3D Model Preparation

The study was conducted on 3D-printed models created after laser scanning of the right tibia with menisci derived from a dog cadaver. Ethical approval was not required for this study according to Polish legal regulations. After a coin toss, the right tibia with menisci was harvested from a fresh cadaver of a 4-year-old adult male golden retriever. The dog died of *Babesia canis* infection and was orthopedically healthy before death. Prior to harvesting the tibia, the mediolateral and craniocaudal radiographs of the right stifle were taken to exclude obvious joint disease. The cruciate ligaments and menisci were found to be intact at the time of the stifle joint disarticulation.

After removing the fibula and soft tissue envelope, the tibia with intact lateral and medial menisci was digitized. A portable HandyScan Black scanner from CREAFORM/AMETEK (Creaform Inc., Levis, QC, Canada), designed to measure 3D objects for inspection and reverse engineering purposes, was used. The scanner emits 7 blue laser crosses that form an automatically generated grid. This allows a fast data transfer from the scanner setup through the digitizing itself to the final file. The bone digitizing process was carried out on a turntable device with reference points (Figure 1). Scanning was performed at a resolution of 0.4 mm with an aperture of 0.35 ms. Other automatic settings to optimize scanning were set to “standard”. Acquisition of the data was carried out in VXelements 11 software (Creaform, Lévis, QC, Canada), whereas processing of the obtained data was carried out in VXmodeler 11 software (Creaform, Lévis, QC, Canada). The scanning process involved taking scans of four sides of the bone (lateral, medial, proximal, and distal) and combining the obtained data into a single digital model. In the next step, automatic filling of surface discontinuities of the obtained model was carried out in the VXmodeler software, and optimization of the generated triangle mesh was carried out in order to obtain the best surface quality. The mesh was optimized by recalculating triangles with a maximum edge length of 0.5 mm and 60° of desired triangle angles. This resulted in an almost uniform distribution of triangles, smooth surfaces, and an optimal number of triangles for further processing in modeling software. The model prepared in this way was exported to an STL (stereolithography) file and then prepared for additive manufacturing technology.

### 2.2. Printing of 3D Models

Digital bone models obtained by 3D scanning were processed by modeling the border of the meniscal plane in order to fill it with resin with a contrasting effect, which allowed accurate delineation of the visible area of the menisci after TPLO. Processing of the models, as well as further preparation for 3D prints, was carried out in Materialise Magics 25.0 processor software, designed to handle a range of 3D printing technologies, and an Ultimaker Cura 5.0 slicer suitable for FDM (Fused Deposition Modeling—a technique of applying very thin layers of molten material to a model from a heated printer nozzle). The bone models were made of white technical thermoplastic polylactic acid (PLA), using a lattice pattern and a fill density of 40%. Then, 15 identical models were printed in 0.2 mm thick layers. Prints were hand-finished. The supports were removed, and the surfaces of the models were sandblasted. Eventually, light-curing red resin was applied to the clearly defined borders of the menisci.

### 2.3. Bone-Holding Device

In order to ensure reliable calculation of the meniscal area before and after TPLO, a self-made bone-holding device was designed. It enabled the positioning of the bone in an unambiguous and reproducible manner in relation to the smartphone camera used to take pictures of the tibial plateau. The holding device was designed in Materialise Magics 25.0 software by performing a logical operation of subtracting the bone from the volume of the cuboid intended to fix the bone firmly. The device was printed the same way as the bone models. The base of the device was made of 8 mm thick laser-cut poly methyl methacrylate (PMMA). A commercially available articulating mount was used to repetitively fix the smartphone against the bone. The bone model was attached to the holding device using a spring-loaded carpenter’s clamp to ensure unambiguous and reproducible positioning in the cuboid volume with the bone outline subtracted. Before and after TPLO, the models were photographed at an angle of 60° to the cranial edge of the tibial plateau (Figure 2 and Figure 3).

### 2.4. TPLO Procedure

Before starting the procedure, a mediolateral calibrated radiograph of one of the 3D models was taken to determine their tibial plateau angle (TPA). Then, the standard TPLO procedure according to the Slocum with jig assistance and semicircular saw R 24 (Iwet vet, Grabówka, Poland) was performed on all 15 models. During this procedure, the rotated proximal tibial segment was stabilized by a TPLO locking plate using three 3.5 mm polyaxial locking screws in the head of the plate and three cortical screws in the body of the plate (Medgal Vet, Księżyno, Poland). Thereafter, mediolateral radiographs of all models were used to determine TPA after TPLO.

### 2.5. Calculation of the Meniscal Visibility

The first step was to take a photo of the menisci of one of the 3D models from a bird’s-eye view. The self-made bone holder was then used to take the images at an angle of 60°, corresponding to the view of the tibial plateau and the menisci highlighted in red before and after TPLO. The photos were then imported into GIMP 2.10.34, a program designed to process graphic images. Because the photos were taken with the same smartphone camera and at the same resolution (Xiaomi 12 Lite, Beijing, China—main camera: triple lenses 108 MP, f/1.9, 26 mm (wide), 1/1.52″, 0.7 µm, PDAF 8 MP, f/2.2, 120° (ultrawide), 1/4.0″, 1.12 µm 2 MP, f/2.4, (macro)), there was no need to calibrate the photographs, as the calculations focused on verifying the change in visibility of menisci in a relative manner, rather than measuring the actual area. In each photograph, the selection by the color tool was used with a sensitivity of 75 (a parameter of tonal variation of surrounding pixels in a relative manner in RGB, the red color was selected from the volume of the resin-marked meniscus area, and then the number of color-marked pixels was read in the histogram tool).

### 2.6. Statistical Analysis

Continuous data (visible part of the entire meniscus area and TPA) did not deviate significantly from the normal distribution (according to the normal probability Q–Q plots and Shapiro–Wilk W test), so they were summarized using the arithmetic mean, standard deviation (±SD), and range. Visible parts of the entire meniscus area were compared between 2 menisci (lateral and medial) and 2 time points (before and after TPLO) using the paired-sample Student’s *t*-test. Differences in visible parts of the entire meniscus area between time points and menisci were presented as the mean difference (±SD) and the 95% confidence intervals (CI 95%). CI 95% values for ratios of visible parts of the entire meniscus area were calculated using the log ratio method. A significance level (α) was set at 0.01 as the Bonferroni adjustment was applied to control for the family-wise error resulting from multiple comparisons (α = 0.05/5 multiple comparisons: 2 comparisons between menisci plus 2 comparisons between time points plus 1 comparison of the increases between menisci) [5]. The analysis was performed in TIBCO Statistica 13.3 (TIBCO Software Inc., Palo Alto, CA, USA).

## 3. Results

The 3D model of the tibia and menisci showed a similar conformation as the natural specimen harvested from the cadaver, maintaining the same length and width, and reproducing the anatomical structures. From the pre- and post-TPLO radiographs, it was possible to identify the anatomical structures corresponding to the tibial plateau.

All TPLO surgeries (n = 15) were performed by the same board-certified surgeon. The preoperative TPA was 26.2°, and the postoperative values ranged between 4.0° and 5.3° (4.6 ± 0.4°) (Figure 4 and Figure 5).

Before TPLO, in the bird’s-eye photo, the total number of red pixels in the lateral and medial menisci was 2,053,995 and 2,140,939, respectively. However, from the surgeon’s perspective, only between 14% and 19% of the entire meniscal surface was visible (Figure 6), and the unhidden meniscal area did not differ significantly between lateral (16.2 ± 1.6%) and medial (16.4 ± 1.6%) menisci (*p* = 0.351). After TPLO, the visible part of the entire meniscus area increased significantly both in the lateral and the medial meniscus (*p* < 0.001)—mean difference ± SD of 30.3 ± 4.3% (CI 95%: 27.9%, 32.6%) and 36.4 ± 6.4% (CI 95%: 32.9%, 40.0%), respectively (Figure 7). However, an observed increase was significantly smaller in the lateral meniscus than in the medial one (mean difference between increases ± SD of −6.2 ± 3.8%; CI 95%: −8.3%, −4.1%; *p* < 0.001). An increase was 2.9-fold (CI 95%: 2.8-fold to 3.0-fold) in the lateral meniscus and 3.2-fold (CI 95%: 3.1-fold to 3.4-fold) in the medial meniscus. As a result, TPLO increased the visibility of the lateral meniscus to 38–56% (mean ± SD: 46.5 ± 5.4%) and the visibility of the medial meniscus to 41–70% (mean ± SD: 52.8 ± 7.6%), and the difference between menisci was significant (*p* < 0.001) (Figure 8).

## 4. Discussion

Our study shows that examination of the menisci after TPLO allows the surgeon to review a roughly threefold bigger area of each meniscus compared to routinely practiced examination before TPLO. It is also the first study that reveals clearly that only a very small part of menisci is visible to the surgeon during classical stifle arthrotomy (roughly 16%). Our technique of examination after TPLO increases the visible part of the menisci to roughly 50%. Even though it is a substantial improvement, roughly half of the meniscus surface remains inaccessible for the surgeon’s sight during stifle arthrotomy.

Before and after TPLO, the menisci were photographed at an angle of 60° to the tibial plateau. This angle was determined by the authors based on their own intraoperative experience. This angle most closely matched the angle at which the authors most often viewed the meniscus after medial mini-arthrotomy and before TPLO. As other surgeons most probably also keep similar positions during this standard approach, it seems that their visibility of both menisci during arthrotomy is limited. Our modeling showed that, in such a case, only approximately 16% of the entire meniscus area was visible (Figure 8). In addition, we calculated that the meniscal visibility can be improved approximately 3-fold by performing their examination after TPLO in which a TPA of approximately 4–5° has been obtained. In such a case, approximately 47% of the lateral meniscus and approximately 53% of the medial one become visible. The clinical benefit appears to be very important as CCL disease more often leads to injuries of the medial than the lateral meniscus [6].

The achieved meniscal visibility after TPLO may depend on the TPA after the surgery. Initially, the TPA of the bone model was 26.2°. Similar values were described in retrievers with intact CCL: 27.97 ± 0.66° [7] and 23.6 ± 3.5° (range 15–29°) [8]. The postoperative TPA is commonly targeted at 5°; therefore, that was the value we chose in our study. Clinically, the ideal TPA usually varies between dogs. Good clinical outcomes have been reported following TPLO with postoperative TPA ranging from 0° to 14° [9]. Most surgeons performing this surgery plan the osteotomy rotation using a chart with an intended postoperative TPA of 5°. We believe that the lower the TPA after TPLO, the better the visibility of the menisci. However, an overrotation of the plateau must be avoided as it may lead to excessive stress on the caudal cruciate ligament and predispose to caudal cruciate ligament injury [10].

The main advantage of our study was that it was performed on 15 identical bone models. This eliminated potential errors in the TPLO technique due to differences in tibial size or shape as well as TPA values.

The main limitation of our study was that we used 3D tibial models instead of stifle joints. The presence of the femoral condyles and other structures can impair the visibility of the menisci. However, this does not influence the results of our study, which showed that the visibility of the menisci improved significantly after TPLO. Therefore, after obtaining good results on 3D models, similar studies should be confirmed on cadavers. In clinical cases, the stifle joint is typically accessed via a craniomedial arthrotomy. Better exposure of the joint will improve the diagnostic accuracy of intraoperative examination and reduce the risk of iatrogenic injury to articular structures. The most challenging aspect of meniscal evaluation is the ability to visualize the caudal pole of the menisci. Many instruments are used to split the articular surfaces of the femur and tibia or to assist in cranial translation of the tibia, e.g., retractors and levers. The surgeon can also manually place the tibia in the cranial drawer to allow visualization of the caudal pole of the lateral or medial meniscus. Based on our clinical experience (unpublished data), we know that it is easier to examine the menisci after splitting the surgical surfaces of the femur and tibia, or after cranial translation of the tibia, if a TPLO has been performed previously.

Nowadays, prototyping and 3D printing offer virtually unlimited possibilities to create even sophisticated anatomical models in a short time [11]. We developed a 3D model of the tibia and menisci to satisfy the need to represent menisci in dogs, as meniscus injury is a very important condition in veterinary medicine. Conducting the laser scans of the tibia with intact menisci was the initial stage of the 3D model production process. In our study, this technique reproduced all anatomical reference points of the canine tibia and menisci. Similar results were achieved by Li et al. [12], who scanned bovine bones to produce 3D models comparable to real bones, and Alcantara et al. [13], who scanned dog pelvis and long bones of the pelvic limbs.

Various types of intra- and extra-articular distractors are used to improve access to the dog menisci. Extra-articular instruments require pins or wire to be inserted into the femur and tibia [2,3,4], while intra-articular distractors during insertion and distraction of the joint can damage intra-articular structures such as cartilage and the caudal cruciate ligament [1]. The increased visibility can improve the accuracy of examination and management of the menisci during arthrotomy. Therefore, we suggest as a routine practice reexamination of the menisci after TPLO, instead of before as it is commonly done [1]. This will reduce the use of joint distractors, making the surgery less traumatic.

## 5. Conclusions

Only roughly 16% of the entire area of menisci can be examined during stifle surgery in dogs if the examination is done before TPLO without additional instrumentation, e.g., arthroscopy. In contrast, examination after TPLO allows the review of roughly 47% of the lateral and roughly 53% of the medial meniscus. Even though it is a substantial improvement, only roughly half of the meniscus surface is still directly visible to the surgeon during arthrotomy. This result strongly indicates that using additional visualizing techniques, such as arthroscopy, should become a routine procedure during CCL disease surgery in dogs.

## Figures and Tables

**Figure 1 animals-14-00065-f001:**
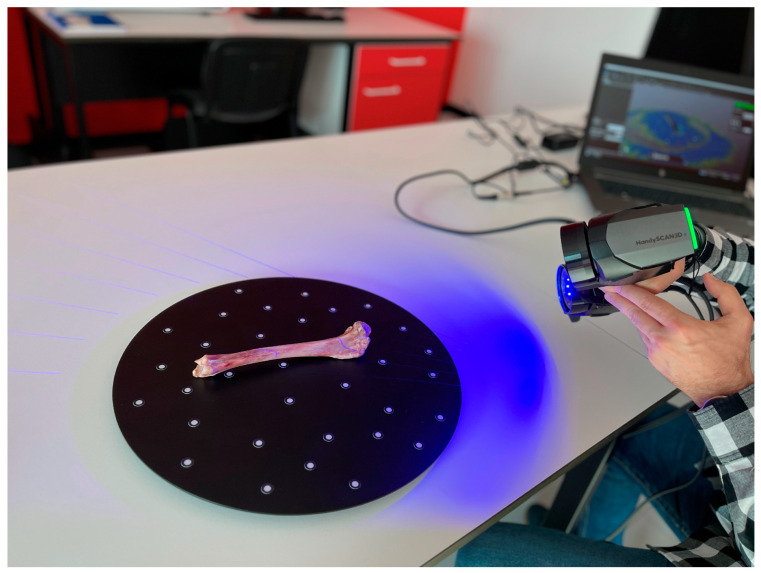
The digitizing process of the right canine tibia with intact menisci.

**Figure 2 animals-14-00065-f002:**
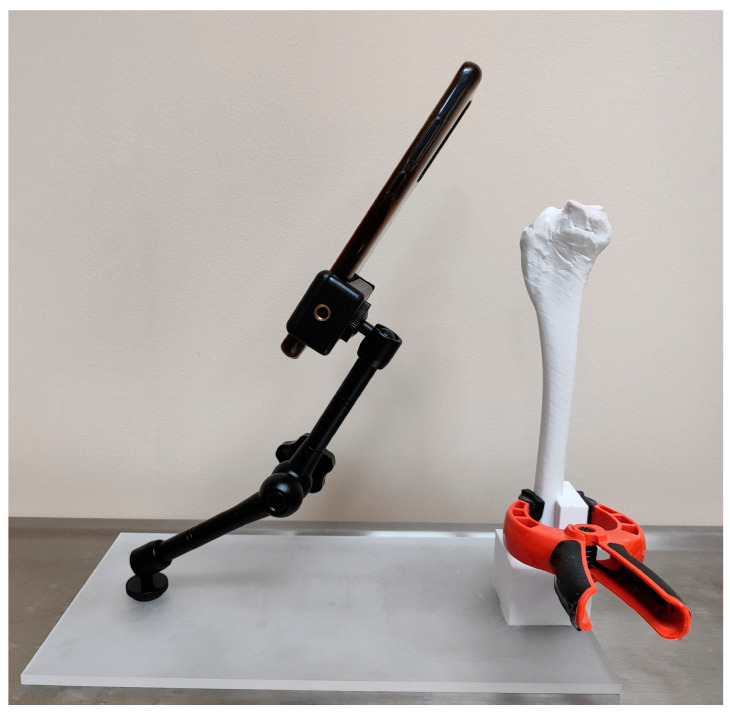
Self-made bone-holding device.

**Figure 3 animals-14-00065-f003:**
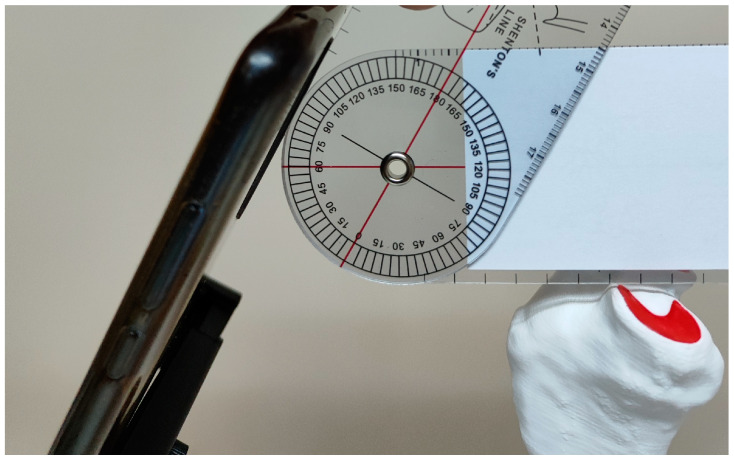
The bone model and camera in a self-made holding device. The model was photographed at an angle of 60° to the cranial edge of the tibial plateau.

**Figure 4 animals-14-00065-f004:**
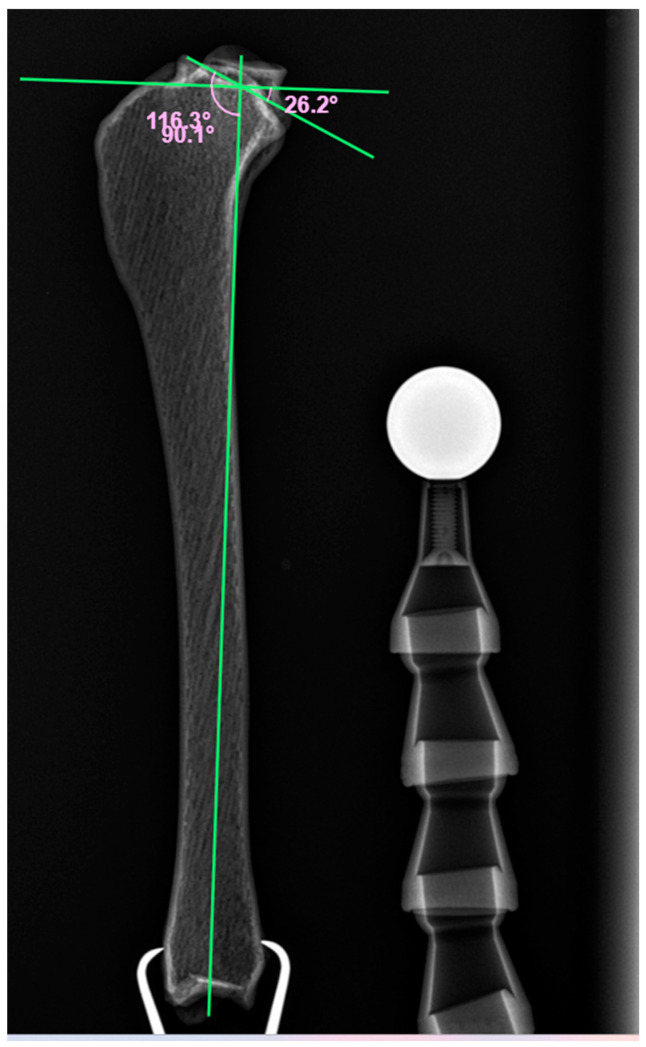
The pre-TPLO X-ray of bone model No. 10.

**Figure 5 animals-14-00065-f005:**
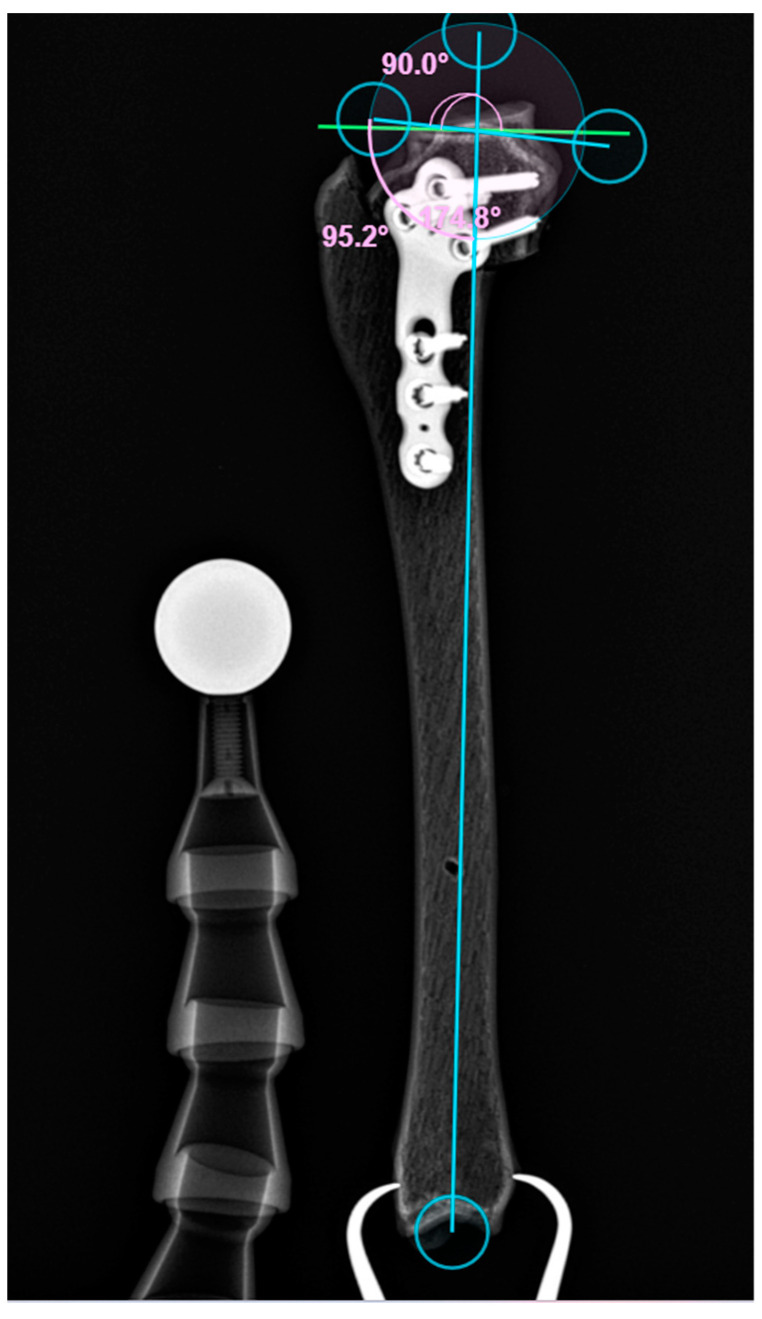
The post-TPLO X-ray of bone model No. 10.

**Figure 6 animals-14-00065-f006:**
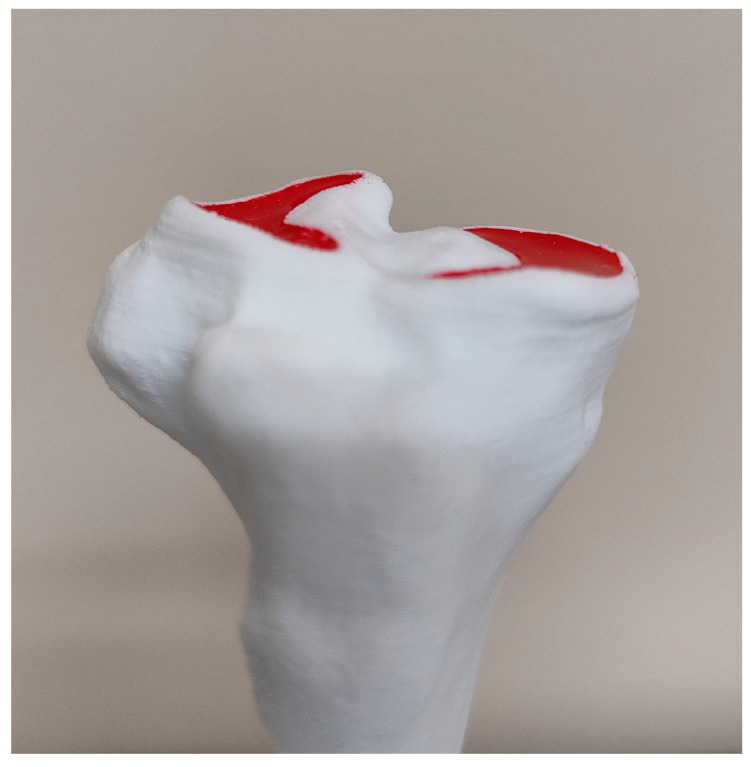
Pre-TPLO photography of the tibial plateau of bone model No. 10. Preoperative tibial plateau angle TPA 26.2°.

**Figure 7 animals-14-00065-f007:**
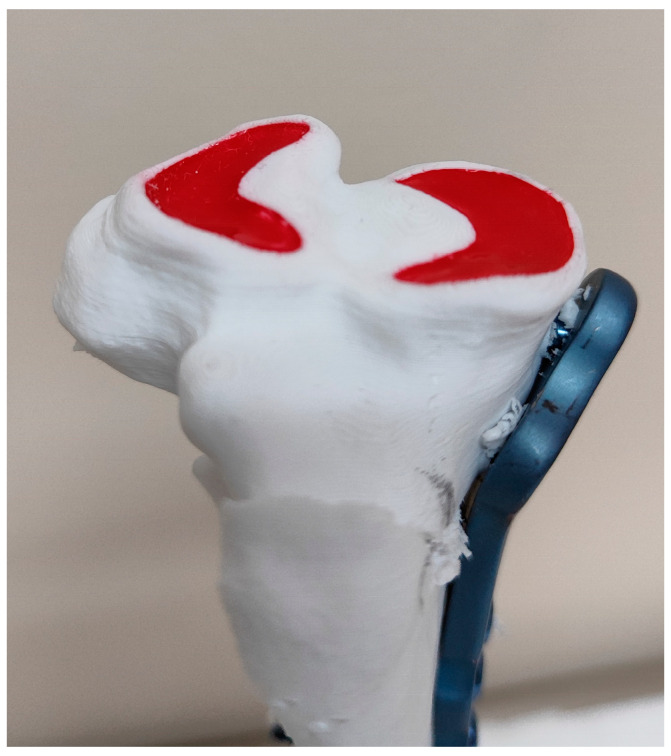
Post-TPLO photography of the tibial plateau of bone model No. 10. Postoperative TPA 5.2°.

**Figure 8 animals-14-00065-f008:**
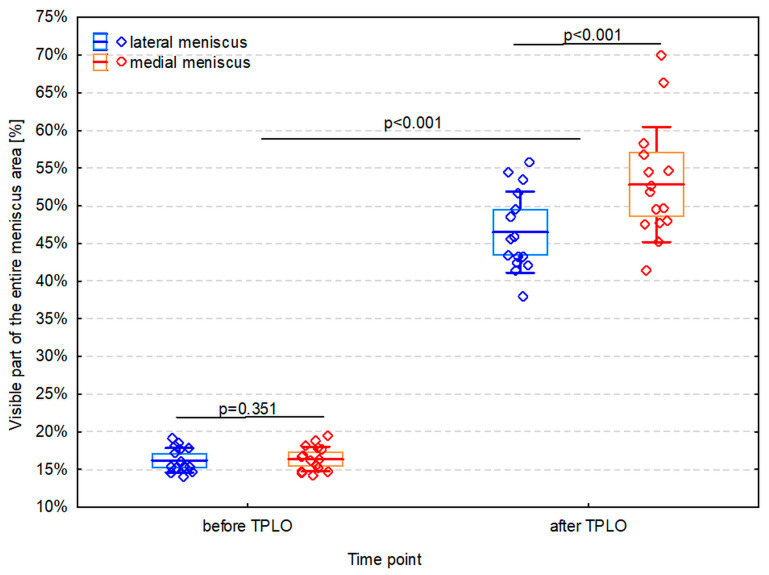
The visible part of the entire area of the meniscus before and after TPLO is presented as the arithmetic mean (line), 95% confidence interval (box), standard deviation (whiskers), and individual observations (diamonds).

## Data Availability

The data presented in this study are available in article.

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
