# Peer review of "Comparison of the Visibility of Canine Menisci before and after Tibial Plateau Leveling Osteotomy: 3D-Printed Model Study"

_animals, 2023, doi:10.3390/ani14010065_

Round 1
Reviewer 1 Report
Comments and Suggestions for Authors
Line 60: space between „curve“ and „of”
Line 89-91: Same statement. Remove one sentence.
Line 105/106: What kind of optimization of the mesh was performed? Smoothing of surface? Re-Scaling of triangle size?
Figure 5: The post TPLO TPA you measured seems not right (negative and overcorrected). I added an orange line, which represents the TPA as I would measure it.
Discussion: You stated, that TPLO leads to improved visualization of the menisci. It seems obvious, that a surface is better visible when it´s turned towards the point of view. However, the effect of the femoral condyle has to be evaluated in the future. When the femur is held in a static position rotating the tibial plateau would hide the caudal horn of the meniscus behind the femoral condyles even more. I would recommend to discuss it a little bit more critical.
Author Response
Dear Reviewer
At first many thanks for the critical review and valuable suggestions. They will definitely improve our paper.
Below the modifications of the manuscript:
Line 60: space between „curve“ and „of”
Answer: we improved
Line 89-91: Same statement. Remove one sentence.
Answer: we remove “At the time of the disarticulation of the stifle, the cruciate ligaments and the menisci were found to be intact.”
Line 105/106: What kind of optimization of the mesh was performed? Smoothing of surface? Re-Scaling of triangle size?
We add Line 118-121
“The mesh was optimized by recalculating triangles with a maximum edge length of 0.5mm and 60° of desired triangle angles. This resulted in an almost uniform distribution of triangles, smooth surfaces and an optimal number of triangles for further processing in modelling software”
Figure 5: The post TPLO TPA you measured seems not right (negative and overcorrected). I added an orange line, which represents the TPA as I would measure it.
Answer: we improved we add correct figure. (as FIGURE 5)
Discussion: You stated, that TPLO leads to improved visualization of the menisci. It seems obvious, that a surface is better visible when it´s turned towards the point of view. However, the effect of the femoral condyle has to be evaluated in the future. When the femur is held in a static position rotating the tibial plateau would hide the caudal horn of the meniscus behind the femoral condyles even more. I would recommend to discuss it a little bit more critical.
We add Line280-290
“In clinical cases, the stifle joint is typically accessed via a craniomedial arthrotomy. A better exposure of the joint will improve diagnostic accuracy of intraoperative examination and reduce the risk of iatrogenic injury to articular structures. The most challenging aspect of meniscal evaluation is the ability to visualize the caudal pole of the menisci. Many instruments are used to split the articular surfaces of the femur and tibia or to assist in cranial translation of the tibia, e.g. retractors and levers. The surgeon can also manually place the tibia in the cranial drawer to allow visualization of the caudal pole of the lateral or medial meniscus. Based on our clinical experience (unpublished data), we know that it is easier to examine the menisci after splitting the surgical surfaces of the femur and tibia or after cranial translation of the tibia if a TPLO has been performed previously”

Reviewer 2 Report
Comments and Suggestions for Authors
This is an innovative manuscript about the use of a 3D printing technique to evaluate the menisci in tibias of dogs. The manuscript only needs some minor corrections as below:
Please explain briefly in the introduction Which are the anatomical structures of the tibial plateau?: Lateral and medial condyles of the tibia, all proximal epiphysis of the tibia…
Lines 33, 45, 78, : change the word “meniscus” by menisci” because is plural
Line 47: In conclusion, the examination…
49: Use words different from those of the title
71-72: Change "proximal tibia” to “proximal extreme of the tibia or proximal epiphysis”
79, 84, 260, 262, 265: change “canine” to “dog” or “canid”
138-162: Please include the reference of the camera (marking, MP, lens) that you used to take the pictures.
263: change the term “hind limbs” to “pelvic limbs”
275: scope. In contrast, …
Comments on the Quality of English LanguageThe quality of English is good.
Author Response
Dear Reviewer
At firs many thanks for the critical review and valuable suggestions. They will definitely improve our paper.
Below the modifications of the manuscript:
This is an innovative manuscript about the use of a 3D printing technique to evaluate the menisci in tibias of dogs. The manuscript only needs some minor corrections as below:
Please explain briefly in the introduction Which are the anatomical structures of the tibial plateau?: Lateral and medial condyles of the tibia, all proximal epiphysis of the
tibia…
Answer: we ad linie 65-70
“The anatomy of the proximal tibia is complex. The articular surface is formed by the lateral and medial condyles. Structures associated with the proximal tibia include the patellar ligament, the lateral and medial menisci, the tibial portions of the cranial and caudal cruciate ligaments, the intermeniscal ligament, the lateral collateral ligament, the medial collateral ligament and the popliteal tendon of origin. In addition, the femorotibial joint space is narrow in the dogs.”
Lines 33, 45, 78, : change the word “meniscus” by menisci” because is plural
Answer: we improved
Line 47: In conclusion, the examination…
Answer: we improved
49: Use words different from those of the title
Answer: we improved “Keywords: cranial cruciate ligament disease, dog, laser scanning, reverse engineering, additive manufacturing, proximal tibial osteotomy”
71-72: Change "proximal tibia” to “proximal extreme of the tibia or proximal epiphysis”
Answer: we improved
79, 84, 260, 262, 265: change “canine” to “dog” or “canid”
Answer: we improved
138-162: Please include the reference of the camera (marking, MP, lens) that you used to take the pictures.
Answer: we ad linie 178-180
“China - main camera: triple lenses 108 MP, f/1.9, 26mm (wide), 1/1.52", 0.7µm, PDAF 8 MP, f/2.2, 120Ëš (ultrawide), 1/4.0", 1.12µm 2 MP, f/2.4, (macro),”
263: change the term “hind limbs” to “pelvic limbs”
Answer: we improved
275: scope. In contrast, …
Answer: we improved

Reviewer 3 Report
Comments and Suggestions for Authors
The Authors presented here the comparison of the degree of visibility of the lateral and medial meniscus before and after TPLO on a 3D printed model of the canine tibia. The results of this study has a clinical aspect and can be helpful especially for the veterinary surgeons. Moreover, the obtained results can serve as an introduction to further research in this area.
I`m sending my suggestion for revision:
- In the abstract the Authors wrote: "In the bird's eye photo, the total number of red pixels in the lateral and medial meniscus was 2053995 and 2140939, respectively" while then in the results section there is an information: "Before TPLO, in the bird's eye photo, the total number of red pixels in the lateral and medial menisci was 2053,995 and 2140,939, respectively." - please make a correction
- Materials and methods - "3. D model preparation" change as: "3D model preparation"
Author Response
Dear Reviewer
At firs many thanks for the valuable suggestions. They will definitely improve our paper.
Below the modifications of the manuscript:
The Authors presented here the comparison of the degree of visibility of the lateral and medial meniscus before and after TPLO on a 3D printed model of the canine tibia. The results of this study has a clinical aspect and can be helpful especially for the veterinary surgeons. Moreover, the obtained results can serve as an introduction to further research in this area.
I`m sending my suggestion for revision:
- In the abstract the Authors wrote: "In the bird's eye photo, the total number of red pixels in the lateral and medial meniscus was 2053995 and 2140939, respectively" while then in the results section there is an information: "Before TPLO, in the bird's eye photo, the total number of red pixels in the lateral and medial menisci was 2053,995 and 2140,939, respectively." - please make a correction
Answer: we improved
- Materials and methods - "3. D model preparation" change as: "3D model preparation"
Answer: we improved
